SOFTWARE

# Facilitating bioinformatics reproducibility with QIIME 2 Provenance Replay

**Christopher R. Keefe**[1], **Matthew R. Dillon**[1], **Elizabeth Gehret**[1], **Chloe Herman**[1,2], **Mary Jewell**[3], **Colin V. Wood**[1], **Evan Bolyen**[1], **J. Gregory Caporaso**[1,2] *

**1** Center for Applied Microbiome Science, Pathogen and Microbiome Institute, Northern Arizona University, Flagstaff, Arizona, United States of America, **2** School of Informatics, Computing and Cyber Systems, Northern Arizona University, Flagstaff, Arizona, United States of America, **3** Department of Epidemiology, University of Washington, Seattle, Washington, United States of America

* greg.caporaso@nau.edu

**Data Availability Statement:** Provenance Replay is open source and free for all use (BSD 3-clause license). The original stand-alone version is available at https://github.com/qiime2/provenance-lib. As of

## Abstract

Study reproducibility is essential to corroborate, build on, and learn from the results of scientific research but is notoriously challenging in bioinformatics, which often involves large data sets and complex analytic workflows involving many different tools. Additionally, many biologists are not trained in how to effectively record their bioinformatics analysis steps to ensure reproducibility, so critical information is often missing. Software tools used in bioinformatics can automate provenance tracking of the results they generate, removing most barriers to bioinformatics reproducibility. Here we present an implementation of that idea, Provenance Replay, a tool for generating new executable code from results generated with the QIIME 2 bioinformatics platform, and discuss considerations for bioinformatics developers who wish to implement similar functionality in their software.

## Introduction

Reproducibility, the ability of a researcher to duplicate the results of a study, is a necessary condition for scientific research to be considered informative and credible [1]. Peer review relies on study documentation to maintain the trustworthiness of scientific research [2–4]. Without comprehensive documentation, reviewers may be unable to verify a study's validity and merit, and other researchers will be unable to interrogate the results or learn from the researchers' approach, limiting the study's value.

The biomedical research community has recently been concerned with a "reproducibility crisis," and several high-profile publications have shown researchers unable to confirm findings of original studies [5,6]. This discussion generally focuses on one type of reproducibility failure: an inability to corroborate a study's results. However, this literature neglects a deeper issue: many studies fail to provide even the minimum necessary documentation to reproduce a study's methodology.

Although there is no standard nomenclature for reproducibility in the literature, existing definitions illustrate the goals of different types of reproducibility. For example, the Turing Way defines research as "Reproducible," "Replicable," "Robust," or "Generalizable" based on

QIIME 2 2023.9 (released 11 October 2023) Provenance Replay is included in the QIIME 2 framework, available at https://github.com/qiime2/qiime2.

**Funding:** This work was funded by NCI ITCR award 1U24CA248454-01 to JGC. The funders had no role in the design, implementation or presentation of the work shared here.

**Competing interests:** The authors have declared that no competing interests exist.

whether a study's results can be repeated using methods and data that are the same as, or different from, the original study [7]. Gundersen and Kjensmo create similar categories in their work on reproducibility, but they define a hierarchy based on the degree of generality [8].

We incorporated these ideas into a hierarchy of reproducible research, where different levels of reproducibility are represented by the classes "Reproducible," "Replicable," "Robust," or "Generalizable" (Fig 1). Under this hierarchy, *generalizable* studies produce findings that are corroborated in other contexts, providing the building blocks for advancing scientific knowledge. Lower degrees of generality allow researchers to validate studies and expand focused work toward generalized conclusions [9].

High quality research documentation is essential to all levels of reproducibility. In bioinformatics, research typically involves large datasets, complex computer software, and analytical procedures with many distinct steps. In order to reproduce such a study, one needs both prospective provenance, the analytic workflow specified as a recipe for data creation, and retrospective provenance, the details of the runtime environment and the resources used for analysis [10].

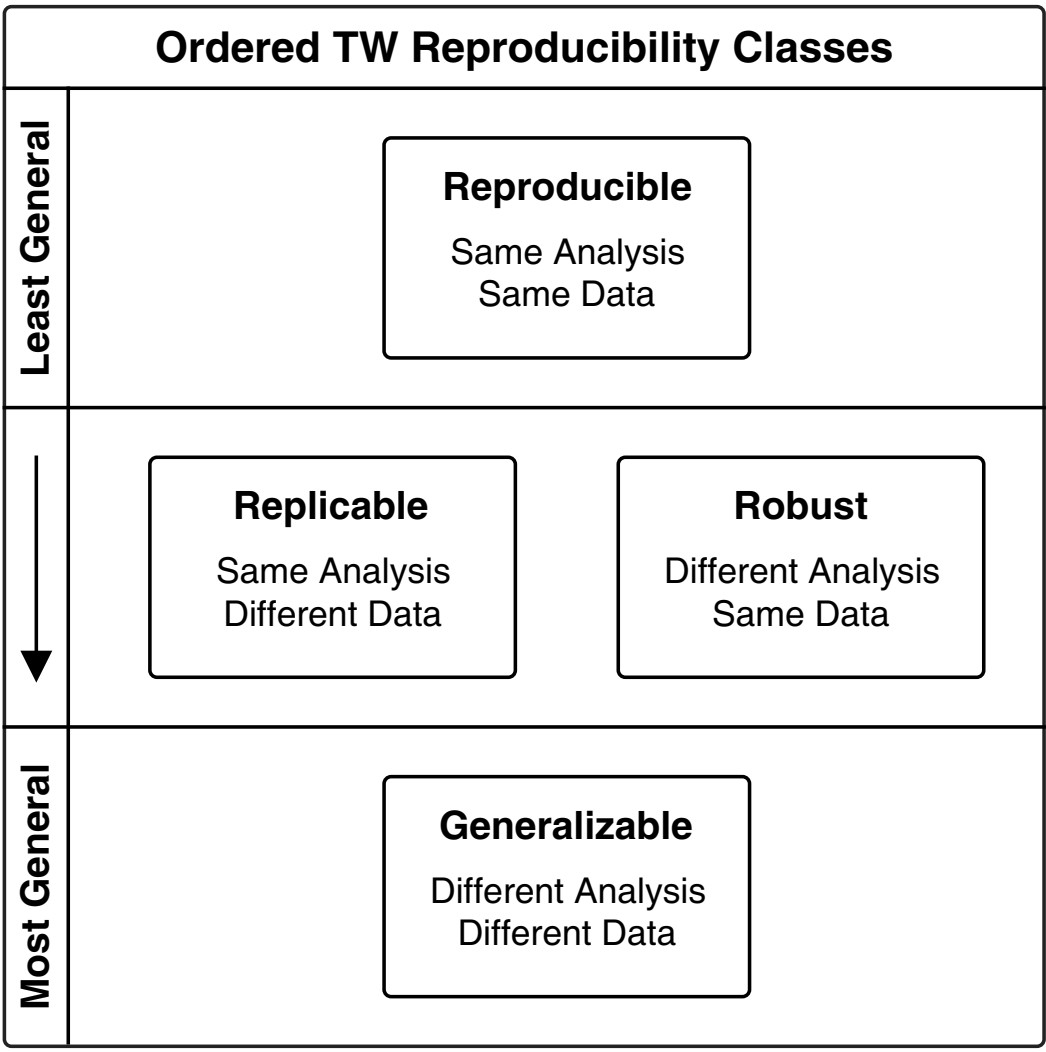

**Fig 1. Turing Way (TW) reproducibility classes [7] ordered in a hierarchy based on generality, similar to Gundersen and Kjensmo [8].**

Prospective provenance is most frequently realized using a document written by the data analyst, for example using Jupyter Notebooks, Snakemake, or RMarkdown documents that include executable code and in some cases analysis notes. Great care must be taken to link revisions of these types of research documentation with the results that they generated, as these documents tend to evolve over time. And even if specific revisions are conclusively linked to specific research results, problems can arise if it's not clear how code was executed in an interactive environment to generate a specific research result. For example, it is not uncommon for novice data scientists to execute cells out of order in a large Jupyter Notebook or RMarkdown file, in which case a linear interpretation of the document would not accurately describe how it was used to generate a result. Retrospective provenance can be realized by capturing information about the analysis as it is run, including hardware and software environments, resource use, and the data and metadata involved. Similar to prospective provenance, this is most frequently captured by a data analyst in the form of written notes. Most published research does not meet these reproducibility needs, as researchers must balance competing demands on their time and grapple with publication structures that incentivize producing new work over documenting for reproducibility [9,11].

Even if researchers know what needs to be tracked and are diligent about tracking that information, recording all of the prospective and retrospective provenance required to reproduce a computational analysis is tedious and error-prone for humans. In our opinion, provenance tracking is a task better left to computer software. Analytic software tools that automatically produce research documentation have the potential to reduce the risk of paper retraction; facilitate collaboration, review, and debugging; and improve the continuity and impact of scientific research [7].

Engineering bioinformatics software to facilitate aspects of analysis reproducibility is a topic of contemporary interest in bioinformatics software literature [12]. For example, Snakemake [13] can automatically generate workflow graphs, using its `--dag` option, providing an automated workflow visualization tool. Love et al [14] present tximeta, an approach for linking reference data checksums to RNA-seq analysis results that rely on that reference data, ensuring that relevant references can be uniquely identified. They also briefly review work highlighting the need for improved provenance tracking in bioinformatics, as well as scientific computing tools that aim to facilitate provenance tracking. Of the existing tools that we are aware of, CWLProv [15] and Research Objects [16] serve the broadest purpose of facilitating reproducibility of entire computational workflows, and are most similar to the work presented here. CWLProv provides a layer between the Common Workflow Language (CWL) and the W3C PROV model, to document retrospective provenance of arbitrary CWL workflows. Research Objects are a concept designed for value-added publication of research products, including research data that is discoverable for other work, and which includes data provenance to enable users to understand how the data was generated.

QIIME 2 is a biological data science platform that was initially built to facilitate microbiome amplicon analysis [17], but has been expanding into new domains, including analysis of highly-multiplexed serology assays [18], pathogen genomics [19], and microbiome shotgun metagenomics, through alternative distributions (i.e., bundles of QIIME 2 plugins, where plugins serve as Python 3 wrappers for arbitrary analytic software, including software written in languages other than Python). QIIME 2 has a built-in system that automatically tracks prospective and retrospective data provenance for users as they run their analyses, and the popularity of this feature is in part responsible for its adoption in other domains. In QIIME 2, users conduct analysis using Actions that each produce one or more Results. The prospective and retrospective provenance of all preceding analysis steps are automatically stored in each Result, allowing users to determine how a Result was generated and an analysis was conducted (Fig 2), even if scripts or

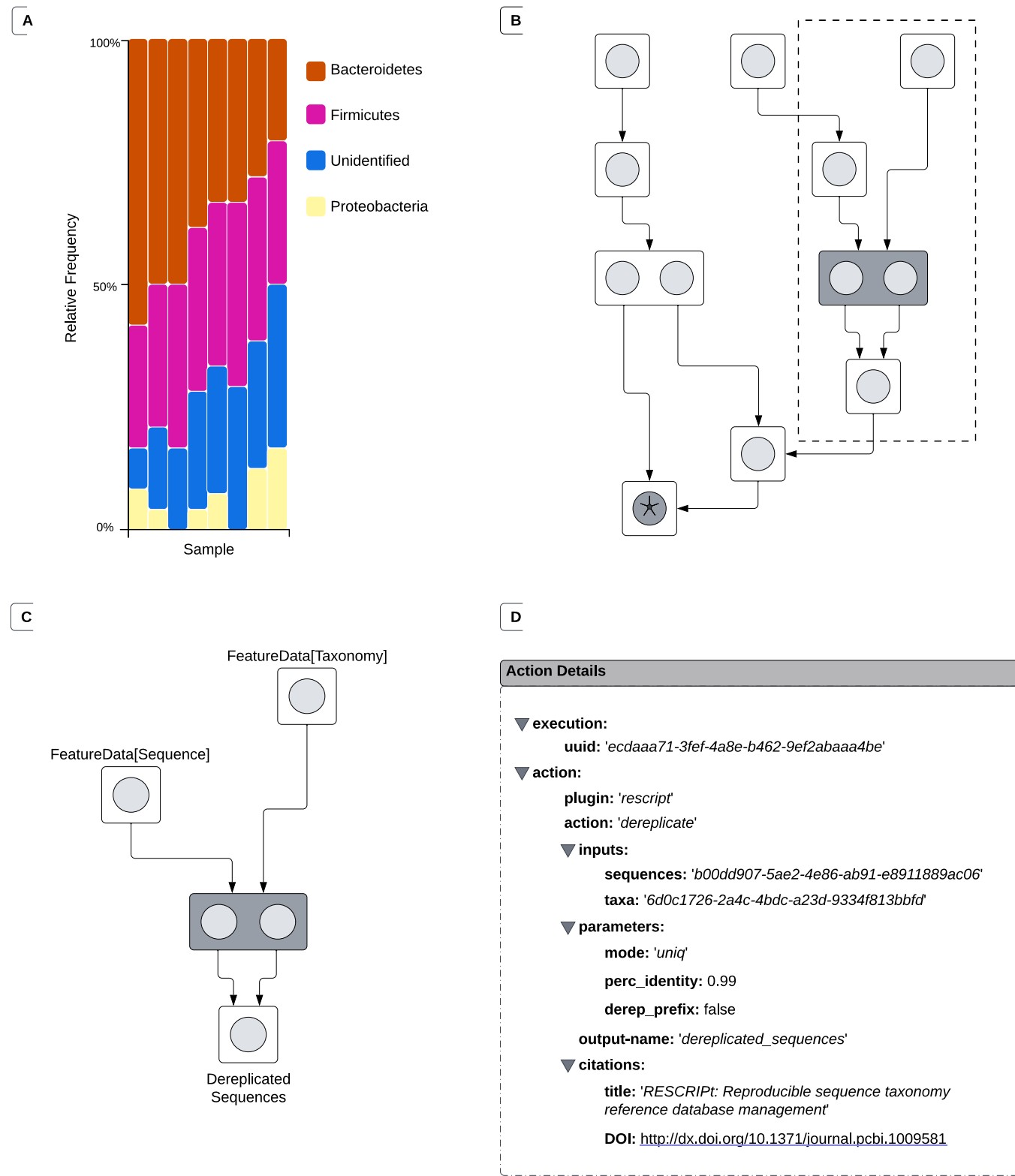

**Fig 2. Schematic diagram of the provenance of a QIIME 2 Visualization.** A: An example of a QIIME 2 visualization illustrating the taxonomic composition of several samples. B: The directed, acyclic graph (DAG) tracing the history of the panel A visualization from initial data import into QIIME 2 through the creation of the visualization (denoted with an asterisk). This DAG can be used for analytical interpretation or publication, and serves as the input to Provenance Replay. Additional detail is provided in panel C on the nodes included in the dashed box. C: A DAG describing the inputs and outputs of the 'action' node highlighted in gray. D: "Action details," captured during the execution of the node highlighted in gray. Some data collected as action details, such as information about the computational environment where the action was run, is not presented in this schematic for readability, but can be observed in the provenance of either of the two QIIME 2 results included in Supporting Information S1. The node selected for panel D was arbitrarily chosen.

notes were not recorded (or were misplaced) by the user, or if revision identifiers of those documents were not linked to research results. Additionally, QIIME 2 assigns universally unique identifiers (UUIDs) to every Result it creates, enabling data to be conclusively identified. Taken together, QIIME 2 therefore automatically supports *reproducible*, *replicable*, and *robust* bioinformatics analyses, without any effort on the part of its users.

The work presented here attempts to improve computational methods reproducibility in bioinformatics by reducing the practical overhead of creating reproducibility documentation. Built around the automated, decentralized provenance capture implemented in QIIME 2, we present Provenance Replay, a tool that validates the integrity of QIIME 2 Results, parses the provenance data they contain, and programmatically generates executable scripts that allow for the reproduction, study, and extension of the source analysis.

## Design and implementation

Provenance Replay is written in Python 3 [20], and depends heavily on the Python standard library, NetworkX [21], pyyaml [22], and QIIME 2 itself, from which it takes advantage especially of the PluginManager and the Usage API. It ingests one or more QIIME 2 results and parses their provenance data into a Directed Acyclic Graph (DAG), implemented as a NetworkX DiGraph. It produces outputs by subsetting and manipulating this DiGraph and its contents. Outputs include BibTeX-formatted [23] citations for all Actions and Plugins used in a computational analysis, and executable scripts targeting the user's preferred QIIME 2 interface. Users interact with the software through a command-line interface implemented with Click [24], or using its Python 3 API.

The initial software design was based on literature review, existing API targets, and discussion with QIIME 2 developers, as well as an initial requirements engineering process. The requirements engineering process consisted of requirements elicitation by focus group and requirements validation using the Technology Acceptance Model (TAM) [25]. Focus group participants were recruited through posts on the QIIME 2 community forum and participated in one-hour focus group sessions, which included a software demonstration and discussion. Discussion questions were intended to elicit open-ended feedback and exploration of possible features of value. When asked how likely they were to recommend Provenance Replay to a colleague who uses QIIME 2, 79% of focus group participants (15/19) were classified as promoters (scores of 9–10 out of 10), 21% (4/19) as passive (scores 7–8 out of 10) and none as detractors, resulting in a net promoter score of +79%. Provenance Replay also scored well on the TAM instruments, with respondents rating both its Perceived Ease of Use as "high," (overall mean 5.8, on a scale of 1–7) and Perceived Usefulness as "high" (overall mean 6.0, on a scale of 1–7). Additional detail on this process is provided in [26].

Provenance Replay is supported by QIIME 2 versions 2021.11 and newer, and can parse data provenance from Results generated with any version of QIIME 2. Provenance Replay is capable of replaying a single QIIME 2 Result in a few seconds, and a very large analysis (450 results) in 8–10 minutes on a contemporary small-business laptop (Intel Core i7-8565U CPU @ 1.8GHz, 16 GB RAM, OpenSUSE Tumbleweed running on an M.2 SSD). As such, most users will not need to work in a cluster environment, and native installation is recommended (and supported on Linux, macOS, and Windows via Windows Subsystem for Linux 2 (WSL2)).

## Results

Provenance Replay is software for the documentation and enactment of *in silico* reproducibility in QIIME 2, which can produce command-line (bash) and Python 3 scripts directly from a

QIIME 2 Result. Provenance Replay outputs are self-documenting, using UUIDs to identify them as products of specific QIIME 2 Results, and they include step-by-step instructions for users to execute the scripts produced. Provenance Replay also implements MD5 checksum-based validation of Result provenance, which can alert if the Results were altered since they were generated, in which case the data provenance would no longer be reliable.

Provenance Replay has many features that we consider good general targets for tools that automate reproducibility documentation:

- Completeness: Provenance Replay provides comprehensive access to captured provenance data.

- Ease of Documentation: Users can generate a complete "reproducibility supplement," including replay scripts and citation information, with a single command through different user interface types.

- Ease of Application: Replay scripts are executable with minimal modification, target a variety of interfaces, and are self-documenting.

- Accessibility: Replay documents are designed for human readability and include their own usage instructions. Additionally, by providing multiple user interfaces for running Provenance Replay, as well as multiple target interfaces for its outputs, users with varying degrees of computational experience can interpret its results.

Provenance Replay automatically removes most barriers to *in silico* methods reproducibility in QIIME 2 (with some exceptions discussed below). This simplifies the process of documenting research, and it has already been used to generate *reproducibility supplements* for scientific publications [27,28].

## Availability and future directions

The Provenance Replay software is open source and free for all use (BSD 3-clause license). As of QIIME 2 2023.5 (released 24 May 2023), the software is included in the QIIME 2 "core distribution" (such that it is installed with QIIME 2), and as of QIIME 2 2023.9 (released 11 October 2023) it is included in the QIIME 2 framework itself, ensuring that it will stay current as QIIME 2 continues to evolve and will be available in all QIIME 2 distributions, including any developed by third-parties.

*Reproducible* and *robust* bioinformatics (Fig 1) involves unambiguous identification of the data used in an analysis, and enabling this is an important target for tools aiming to facilitate reproducible bioinformatics. This can be challenging to achieve, as it requires stable, unique identifiers, and if data can be mutated, the identifiers should be versioned. QIIME 2 uniquely identifies its data artifacts with UUIDs, and those artifacts are immutable (once created, they cannot be changed without the creation of a new data artifact with a different UUID). This ensures that analyses are *reproducible* and *robust* if a researcher has access to the data. Providing general purpose access to QIIME 2 Results is outside the scope of the system, so it remains the responsibility of the user to ensure their data are available to others. Sharing QIIME 2 Results and Provenance Replay reproducibility supplements in a single archive through a stable service such as FigShare is an excellent way for users to ensure that their analysis will be *reproducible*, *replicable*, and *robust*. QIIME 2 may facilitate data access in the future by enabling programmatic retrieval of data artifacts from Qiita [29], or integrating q2-fondue [30] commands with Provenance Replay results, to load data from the NCBI Sequence Read Archive into QIIME 2 as a step in replaying an analysis.

In addition to enabling unambiguous identification of data (including any reference data) that is used, there are several other important targets for tools aiming to facilitate reproducible bioinformatics. First, commands used to generate data must be recorded with all parameter settings, including default values, and unambiguously linked to their input and output data. While checksums of entire files are tempting unique identifiers, they take too long to compute on large files to be practical as identifiers. QIIME 2 uses version 4 UUIDs. Next, all relevant software versions, including versions of underlying dependencies, must be recorded. Details about the environment where command execution occurred are important to record, including the Operating System and its version and the version of the programming languages used. Differences in any of these variables could cause a failure to reproduce results. Minting and recording an execution identifier (e.g., as a UUID) can help to remove ambiguity regarding when or how a command or workflow was applied. And, while not essential for reproducibility, recording a timestamp of execution can be helpful when trying to make sense of collections of results. Finally, it is generally a good idea to optionally provide software tools in a containerized environment, to ensure that a working environment will be accessible in the future (e.g., if unavailability of compatible binaries prevents environment recreation). Khan et al. (2019) [15] provide a more detailed list of specific recommendations and corresponding justifications on best practices for ensuring reproducible computational workflows.

A future goal is to enable Provenance Replay to output replay scripts for users of QIIME 2 through graphical interfaces. QIIME 2 can be accessed through a Python 3 API, a command line interface (CLI), and through various workflow systems, including Galaxy and CWL. At present, Provenance Replay can output Python 3 scripts and bash scripts, providing documentation options for users of the API and CLI. A future development target is to provide reproducibility instructions for Galaxy [31] users as well. Providing complete documentation through higher-level (e.g., graphical) interfaces is more verbose, but ultimately expands the audience who can learn from that documentation.

Comprehensive study documentation is a necessary prerequisite to scientific reproducibility, but many researchers are unable to provide adequate documentation due to limited training, resources, and competing demands for their time. Tools such as Provenance Replay provide a means for ensuring study reproducibility while reducing the documentation burden on bioinformatics users, who may forget to record steps in their computational lab notebooks, or who may not be aware of all of the information that needs to be documented to ensure reproducibility. Provenance Replay largely automates *in silico* reproducibility in QIIME 2, and this approach can provide a model for other scientific computing platforms. Moving forward, computational tools that record data provenance for the user will be a major advancement for methods reproducibility, allowing researchers to more easily corroborate results, learn from the work of others, and build on the conclusions of scientific studies.

## Supporting information

**S1 File. qiime2-provenance-replay-code-and-tutorial.** The provenance replay code, as of QIIME 2 2023.9, and a brief usage tutorial with corresponding data. The two .qzv files included in this supplement are "QIIME Zipped Visualization" files. These can be used with QIIME 2 Provenance Replay to generate replay scripts, can be viewed using QIIME 2 View (https://view.qiime2.org), or can be unzipped with any typical unzip utility as they are .zip files with a specific internal structure that enables QIIME 2 to interpret them.
(ZIP)

## Author Contributions

**Conceptualization:** Christopher R. Keefe, Matthew R. Dillon, Evan Bolyen, J. Gregory Caporaso.

**Funding acquisition:** J. Gregory Caporaso.

**Investigation:** Christopher R. Keefe.

**Software:** Christopher R. Keefe, Colin V. Wood.

**Supervision:** Matthew R. Dillon, Evan Bolyen, J. Gregory Caporaso.

**Validation:** Chloe Herman.

**Visualization:** Elizabeth Gehret.

**Writing – original draft:** Christopher R. Keefe, Mary Jewell.

**Writing – review & editing:** Christopher R. Keefe, Matthew R. Dillon, Elizabeth Gehret, Chloe Herman, Mary Jewell, Colin V. Wood, Evan Bolyen, J. Gregory Caporaso.

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
