## [Decision Letter · Decision Letter 0]

19 Sep 2023

Dear Dr Caporaso,

Thank you very much for submitting your manuscript "Facilitating Bioinformatics Reproducibility with QIIME 2 Provenance Replay" for consideration at PLOS Computational Biology.

As with all papers reviewed by the journal, your manuscript was reviewed by members of the editorial board and by several independent reviewers. In light of the reviews (below this email), we would like to invite the resubmission of a significantly-revised version that takes into account the reviewers' comments.

Dear Authors,

Thank you for submitting your manuscript to PLoS Computational Biology. It has now been reviewed by two experts who have done a thorough job and made constructive suggestions for improvement. I have also read the manuscript and agree with their assessment and recommendations.

I agree it will be important to reduce the reliance on the master's thesis by going back to the original citations and this would also be fair to the authors of these studies.

Regarding the tension between developing the concepts about reproducibility and explaining the actual work sufficiently, I agree that the latter is essential but if the authors can manage to better explain the conceptual work in a concise way, I would be happy to have that included. However it must become clearer than it is at the moment. I felt the explanation was not clear enough for someone not familiar with the ideas already and for those who already are, it is less of a problem but not doing a great job as to what are new ideas and why they are important. If you decide to develop this into a separate publication that is fine. If you want to keep it, make it clearer while keeping it concise.

Best wishes,

Jan-Ulrich Kreft

Guest Editor

We cannot make any decision about publication until we have seen the revised manuscript and your response to the reviewers' comments. Your revised manuscript is also likely to be sent to reviewers for further evaluation.

Sincerely,

Jan-Ulrich Kreft

Guest Editor

PLOS Computational Biology

Mark Alber

Section Editor

PLOS Computational Biology

Dear Authors,

Thank you for submitting your manuscript to PLoS Computational Biology. It has now been reviewed by two experts who have done a thorough job and made constructive suggestions for improvement. I have also read the manuscript and agree with their assessment and recommendations.

I agree it will be important to reduce the reliance on the master's thesis by going back to the original citations and this would also be fair to the authors of these studies.

Regarding the tension between developing the concepts about reproducibility and explaining the actual work sufficiently, I agree that the latter is essential but if the authors can manage to better explain the conceptual work in a concise way, I would be happy to have that included. However it must become clearer than it is at the moment. I felt the explanation was not clear enough for someone not familiar with the ideas already and for those who already are, it is less of a problem but not doing a great job as to what are new ideas and why they are important. If you decide to develop this into a separate publication that is fine. If you want to keep it, make it clearer while keeping it concise.

Best wishes,

Jan-Ulrich Kreft

Guest Editor

Reviewer's Responses to Questions

**Comments to the Authors:**

Reviewer #1: The presented software article motivates and presents a python application capable of extracting metadata from a QIIME2 analysis. This metadata serves as documentation of the analyses performed within QIIME2, and can be used to re-run the analyses. The advantage of programmatically extracting this documentation, or data provenance information, is that it requires very little time and effort from the researcher and guarantees retrieval of accurate and complete information, which can be tedious to achieve manually.

The application is a useful and much needed addition to the QIIME2 software suite, making it easier and faster for researchers to provide provenance information in scientific publications and related works. The idea is not particular novel, and references to exiting work need to be added, but means to increase the chances that scientists provide good provenance data are very important.

The presentation of the application, its output, and the placement in the field should be improved before publication as outlined in the major comments below.

Major comments:

- The last sentence of the abstract says ‘… and discuss considerations for bioinformatics developers who wish to implement similar functionality in their software.’ I cannot find this back in the Results or Future Directions sections. I think it would be useful to list the key elements that should be reported for microbiome research, and any suggestions you might have for developers in the field.

- The manuscript heavily relies on the reference Keefe 2022, a master thesis which is not easily accessible, only upon request. I feel this is a major limitation and all information necessary to fully understand the paper, its motivation and the application should be given in the manuscript or supplement.

- No proper introduction of the field of workflow documentation is given, and references to similar approaches are missing. This needs to be added. For example, Snakemake generates an analysis graph (DAG) and provides documentation during runtime, the config file allows ‘replay’ at any time. Other post-analysis documentation tools exist but are not discussed and cited. See how Love et al, 2020, provided an overview over existing applications for provenance tracking of RNA-seq data for how it should look like.

- I like the reference to the reproducibility classes of the Turing Way categories/Gundersen. The figure legend (figure 1) is a bit hard to digest though. The concept is clear and easy to understand, so make the figure legend easy to comprehend as well, cite the references in the legend as well, and keep the details for the main text.

- The authors do not place their application in the context of Figure 1. Please do so and discuss the challenges associated with the more general reproducibility classes.

- Figure 2 is extremely difficult to interpret with the limited information given. Even if the reader is very familiar with the steps of the analysis, it is impossible to match input and output data and actions/methods to the DAG. The DAG needs to be sensibly annotated to be useful. Same holds for the Action details which are almost not human-readable in the presented form. Make sure the action details are easily matched to the DAG and the input/output files of the user, to make the DAG a useful feature of Provenance Replay.

Minor comments:

- Avoid spoken expressions like “aren’t” in the text (abstract).

References:

[Love et al., 2020] Love MI, Soneson C, Hickey PF, Johnson LK, Pierce NT, et al. (2020) Tximeta: Reference sequence checksums for provenance identification in RNA-seq. PLOS Computational Biology 16(2): e1007664. https://doi.org/10.1371/journal.pcbi.1007664

Reviewer #2: Keefe et al have developed a useful software component that will, as advertised, serve as an aid to reproducibility in microbiome data analysis. The problem that the authors identify is real: even sophisticated bioinformatics scientists have trouble recording and documenting complex analytical workflows. The solution presented by the authors is to let the computer record the steps and software dependencies from the analysis, a task which seems to be so much more appropriate for computers than humans. While this collusion only covers one field of research on one bioinformatics platform, it does point the way forward for other areas of research and software.

Major comments

1. There is a tension within the article between developing broad ideas about reproducibility vs introducing the software. As the primary purpose of this article is to introduce new software, the authors should not develop broad ideas that are not absolutely necessary to motivate and describe the software presented. As just one example, is it necessary to merge reproducibility categories from two organizations into a unified hierarchy? Can’t the software’s utility be justified separately in the context of each framework, without merging them? Personally, I think the broad ideas are interesting and would make for an impactful, but separate, manuscript. The authors do not have the space to properly develop broad ideas about reproducibility here, and any attempts to do so will distract from the main focus of the paper.

2. Much of the background for this manuscript seems to come from a Master’s thesis written by the lead author. The thesis is primarily used as a source for the goals that the software seeks to meet. I tried but was unable to obtain a copy of the thesis from the library at Northern Arizona University. I understand referencing a thesis, but am uncomfortable with the degree to which the basis of the article relies on a thesis from the lead author. As I’ve not read the thesis, I’m not sure about the degree to which the goals are crafted by the author himself vs. aggregated from other sources. If the goals outlined in O’Keefe 2022 are collected from other sources, please cite them directly. Throughout the article, the reliance on O’Keefe 2022 should be minimized in favor of briefly describing the rationale and citing original sources.

3. In the Design and Implementation section, the authors say, “initial software design was based on literature review, existing API targets, and discussion with QIIME 2 developers, as well as an initial requirements engineering process and formal focus groups with prospective users.” Even if you don’t have space for all the details, we at least need to hear about the requirements engineering process, and we need to see some data from the focus groups. Because you have gone above and beyond the design practice in many labs, the readers need to see what a robust design process looks like. And if you did it right, these data will provide strong support for your design.

4. In Availability and Future Directions, the authors say, “this approach can provide a model for other scientific computing platforms,” but then don’t address HOW other computing platforms might use ideas from QIIME 2 Provenance Replay. This is the one glaring question that needs to be addressed by the authors. It seems to me that the design was so successful in QIIME 2 because the framework was built from the ground up to keep track of software used in each step. Is the plan to decouple QIIME 2 from microbiome data analysis and extend into other areas? To have separate QIIME 2-like frameworks for each field of research? To develop a general tool that tracks software dependencies on the command line? How does Galaxy fit into this picture? Do Jupyter Notebooks and RMarkdown have anything to learn from this? If you have to trim other parts of the article to make space for exploring this question, please do. This is where you can emphasize the payoff from your software and examine some broad ideas while staying relevant to the topic of the article.

Minor comments

1. The reproducibility hierarchy introduced by the authors, merging work from Gunderson and Kjensmo with ideas from the Turing Way, is presented in an overly technical way. Instead of “we contextualize key factors in generalizability within the broader goals of reproducible research,” you could just as easily say “we developed an expanded hierarchy that includes ideas from both groups.” The figure legend is impossible to understand without a full understanding of the text, which is itself difficult. This prevents the figure from serving as a visual introduction to your ideas about reproducibility. This may be a moot point if you follow the suggestion from major comment 1.

2. You probably mean to cite the Turing Way as “The Turing Way Community. (2021, November 10). The Turing Way: A handbook for reproducible, ethical and collaborative research. Zenodo. http://doi.org/10.5281/zenodo.3233853”

**Have the authors made all data and (if applicable) computational code underlying the findings in their manuscript fully available?**

Reviewer #1: Yes

Reviewer #2: Yes

PLOS authors have the option to publish the peer review history of their article (what does this mean?). If published, this will include your full peer review and any attached files.

Reviewer #1: **Yes: **Julia C Engelmann

Reviewer #2: No
---

## [Editor Report · Decision Letter 1]

10 Nov 2023

Dear Dr Caporaso,

Thank you for the thorough revision addressing all comments raised by the reviewers satisfactorily. There is therefore no need to send it out to the reviewers again and we are pleased to inform you that your manuscript 'Facilitating Bioinformatics Reproducibility with QIIME 2 Provenance Replay' has been provisionally accepted for publication in PLOS Computational Biology.

There are a few minor changes that I would like you to consider at the proofing stage. In the Design and Implementation section, you report percentages of focus group participants with 4 significant figures although there were only 19. The values should be rounded to two sig. fig. Reporting the TAM sores of e.g. 5.82 would be easier to interpret if you stated the maximum score. In Results you use the term computational literacy, it may be nicer to use experience instead.

Best regards,

Jan-Ulrich Kreft

Guest Editor

PLOS Computational Biology

Mark Alber

Section Editor

PLOS Computational Biology

---

## [Editor Report · Acceptance letter]

20 Nov 2023

PCOMPBIOL-D-23-00967R1 

Facilitating Bioinformatics Reproducibility with QIIME 2 Provenance Replay

Dear Dr Caporaso,

I am pleased to inform you that your manuscript has been formally accepted for publication in PLOS Computational Biology. Your manuscript is now with our production department and you will be notified of the publication date in due course.

With kind regards,

Anita Estes
